# Deep hierarchical model for hierarchical selective classification and zero shot learning

## Abstract

Object recognition in real-world image scenes is still an open problem. With the growing number of classes, the similarity structures between them become complex and the distinction between classes blurs, which makes the classification problem particularly challenging. Standard N-way discrete classifiers treat all classes as disconnected and unrelated, and therefore unable to learn from their semantic relationships. In this work, we present a hierarchical inter-class relationship model and train it using a newly proposed probability-based loss function. Our hierarchical model provides significantly better semantic generalization ability compared to a regular N-way classifier. We further proposed an algorithm where given a probabilistic classification model it can return the input corresponding super-group based on classes hierarchy without any further learning. We deploy it in two scenarios in which super-group retrieval can be useful. The first one, selective classification, deals with the problem of low-confidence classification, wherein a model is unable to make a successful exact classification. The second, zero-shot learning problem deals with making reasonable inferences on novel classes. Extensive experiments with the two scenarios show that our proposed hierarchical model yields more accurate and meaningful super-class predictions compared to a regular N-way classifier because of its significantly better semantic generalization ability.

## 1 Introduction

Object recognition from images in real world scene, which requires drawing boundaries between groups of objects in a seemingly continuous space, is a highly complex problem. The visual world is populated with a vast number of diverse objects, where each object instance may pose many visual forms. Furthermore, with the growing number of classes, the similarity structures between them become complex and the distinction between classes blurs. Because of the high complexity of this task, most existing work focus on simplifying the problem to a supervised single-label classification problem. Unfortunately, since one-vs-all classifiers treat all classes as unrelated, visual recognition systems cannot transfer semantic information about learned classes.

One way of dealing with this issue is to respect the natural continuity of visual space instead of artificially partitioning it into disjoint categories. Classes have varying degrees of discriminability. Some classes have unique features while other classes might share similar features and are hence harder to distinguish from each other. Such similarity structures in the data are very valuable information that could potentially lead to improving classifiers (Frome et al., 2013; Rippel et al., 2015; Bilal et al., 2018). Such classifier has a potential to make semantically reasonable errors. For example, in the upper image of Figure 1 the standard model choose the wrong class 'stingray' while a more reasonable error can be 'hammerhead' or 'great white shark'. Moreover, such semantic classifiers may be less biased to irrelevant context as illustrated in the example. The standard model is biased to the wrong class because the image has people. We found that in the training set many images of stingray are with people. In order to confirm this statement we occlude[1] the people. By doing this we mange to classify the image correctly with the standard classifier (Zeiler & Fergus, 2014). On the other hand, our semantic model identify the true class even in the original image. Furthermore, after

---

[1]Occlusion technique is generally used to reveal the regions which contribute most to a target response, here we used it to identify the error cause

occluding the error cause our model ranks the top-5 predicated classes according to their relevance to the true class measured by the taxonomy distance.

We propose an approach that addresses these shortcomings by learning a hierarchical inter-class relationship model. We present a novel hierarchical probability-based loss function, which we call **soft-NLL**. This loss function gives a probability weight according to the inter-class taxonomy graph distances. We evaluate the model semantic generalization ability, which means for example that the model makes more semantically reasonable errors. Our model shows a significant improvement on the hierarchical semantic metric in an order of magnitude over existing methods, while it guarantees a minor decrease in the common top-k accuracy compared to the widely used hard-NLL loss. Our approach is independent of the specific prediction model and thus can always benefit from design progress.

Furthermore, we develop an algorithm where given a probabilistic model, based on its top-k predictions our algorithm returns the input corresponding super-group based on classes hierarchy without any further learning. We present two use cases for super-group retrieval and show that our soft-NLL outperforms in both applications. The first one, selective classification, deals with the problem of low-confidence classification, wherein a model is unable to make a successful exact classification. In this case, our algorithm returns a corresponding closest super-class. In the second scenario, the proposed method is used for the zero-shot learning problem. In this case, given a novel input, the algorithm returns its hierarchically related group, rather than generating a true unseen group.

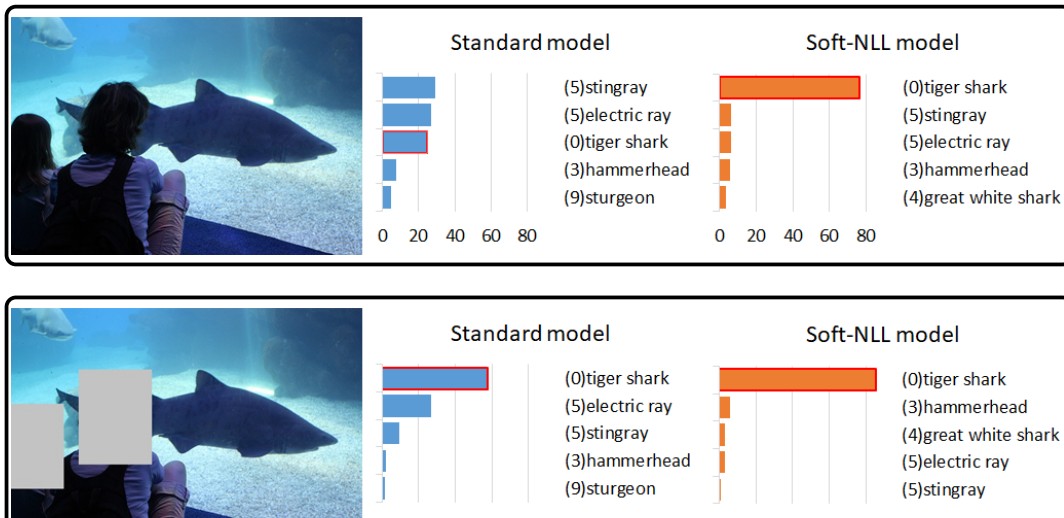

Figure 1: Comparison example of our model and standard classifier top-5 predictions softmax responses taken from ILSVRC2012 validation set, where the true class is 'Tiger shark'. The number next to class name indicates WordNet taxonomy distance from the true class calculated to that class

## 2 RELATED WORK

Vision researchers have sought to exploit external semantic knowledge. Taxonomies, or class hierarchies were the first to be explored (Griffin & Perona, 2008; Marszalek et al., 2008). Taxonomy depends on a human-defined criteria, and thus implicitly consists of semantic knowledge. Early works (Bengio et al., 2010; Gao & Koller, 2011; Griffin & Perona, 2008; Marszalek et al., 2008) exploit the tree structure for efficient branch-and-bound training and prediction, while a few use taxonomies as sources of relational knowledge between categories (Deng et al., 2014; Hwang, 2013). Our method shares the same goal with the latter group of work.

Deep Convolutional Neural Networks which learn high-level features and a N-way flat softmax classifier have recently gained enormous popularity for their impressive performance on a number of visual recognition tasks. Since Krizhevsky et al. (2012) won the ImageNet ILSVRC challenge

2012 (Deng et al., 2009), many variants of this model have been proposed. GoogLeNet (Szegedy et al., 2015), Resnet (He et al., 2015) and VGGNet (Simonyan & Zisserman, 2014) focus on increasing the network depth by adding more convolutional layers to the original model. These models generalize better than smaller ones. Our goal is to benefit form the progress of CNN designs.

Some existing work has explored various approaches to learning semantic information in deep neural networks. One related approach is to learn a joint representation of images and labels in an embedding space. Frome et al. (2013); Norouzi et al. (2013) learn semantic relationships between classes using a semantic word embedding model, where each class represented by an embedding vector. In parallel, they pre-train a deep neural network for visual object recognition. Then, both methods develop a specific mapping of the images into the semantic embedding space. Both methods evaluated their semantic ability using a semantic metric. Moreover, they employed their models for zero-shot learning. However, Frome et al. (2013) show modest performance in its semantic relevance meaning because the word2vec embeddings are weak in the sense of capturing the inherent hierarchical semantics. Furthermore, it imposes structure only on the output space, and structure on the learned space is not explicitly enforced. However, our goal is to transfer direct inter-class hierarchical data to the learned space.

Deng et al. (2014) introduce a new formalism which captures semantic relations between any two labels applied to the same object:mutual exclusion, overlap and subsumption. In the experiments part they used Wordnet heirachy and defined exclusion relation between any two labels unless they share a descendant. Each image got a single label and in this scope some leaf examples was relabeled only to theirs immediate parent(s) internal node label. In different to this method, we propose to formalize real mutual exclusion and subsumption relations in a continuous similarity meter calculated from taxonomy. The similarity level of labels is getting lower according to their distance from the true class. We show that their exclusion approach is too strict. That is we show that giving similarity weight even to labels which share predecessor is beneficial to performance in Section 4.2. Secondly, our similarity concept which gives a soften-label allow to share similarity structure on all images. Moreover, the authors did not present an evaluation of how predictions (i.e. even the incorrect ones) are semantically meaningful which is our focus in this work.

Yan et al. (2015) propose a two-staged CNN architecture named HD-CNN, which leverages the taxonomy to separate the categories into easy coarse-grained ones and confusing fine-grained ones, trained in separate networks. However, such separate learning for coarse and fine-grained categories results in larger memory footprints, while it improves performance with only a modest rate. Recently Bilal et al. (2018) designed a hierarchy-aware CNNs that accelerate model convergence and alleviate overfitting. They select AlexNet (Krizhevsky et al., 2012) as a reference architecture and created branches from multilevel deep features to perform group-level classification and back-propagate group error. The number of groups is gradually increased along the depth of the inner layers. However this scheme demands extensive analysis and consequently it cannot easily benefit from the progress of CNN designs. Moreover, the latter two works did not present an evaluation of how predictions are semantically meaningful.

Our goals are to directly exploit the class hierarchy information while benefit form the progress of CNN designs. We focus on large scale multi-class problems. In contrast to most previous works we evaluate the semantic ability of our model according to a semantic metric. We further show that such a metric is meaningful when we compare our soft semantic model to a regular hard-model on two applications.

## 3 THE PROPOSED METHOD

### 3.1 SOFT NEGATIVE LOG LIKELIHOOD LOSS FORMULATION

The problem we address in this paper is a supervised multiclass classification. Following common setup, we assume the input $X \in \mathcal{X}$ and label $Y \in \mathcal{Y} = \{1, ..., \mathcal{C}\}$ can be modelled by a joint distribution $\pi(X, Y)$. The labels are organized in a taxonomy graph $G$, where each directed edge represents an 'is a' relation, and the distance between nodes reflects a semantic relationship between them. Our goal is to model these semantic relationships, so that its top-k predictions will be closely related to the true label according to $G$.

The standard objective formulation used in the context of deep learning for a multiclass classification problem is the negative log likelihood (NLL) loss, which is also referred to as the cross entropy loss with the probability class indicator. Given a probabilistic model $\hat{\pi}(Y|X)$ and a sample $(x_0, y_0)$, NLL loss is defined by

$$l(x_0, y_0) = -\sum_{y=1}^{\mathcal{C}} \pi(y|x_0) \log \hat{\pi}(y|x_0) \text{ , where } \pi(y|x_0) = \mathbf{1}_{(y=y_0)}$$

This formulation assigns the whole probability weight to a single class with all other classes having zero weight. This approach artificially partitions the visual space into disjoint categories and does not take into account semantic relationships between classes.

We propose to use a 'soft' (instead of 'hard') model $\pi(y|x)$ as a semantic relationship measure. This model can be learned as the class graph taxonomy $G$. More formally, let $\Pi \in \mathbb{R}^{\mathcal{C} \times \mathcal{C}}$ represent a semantic probability taxonomy relationship (or distance) map, where $\Pi_{i,j} \triangleq \pi(i|j)$. We define the soft NLL loss as:

$$l(x_0, y_0) = -\sum_{y=1}^{\mathcal{C}} \Pi_{y_0, y} \log \hat{\pi}(y|x_0) \tag{1}$$

$\Pi$ is a class row-wise i.e. each ith row resembles the ith true-class probabilities, which are inversely proportional to the on-the-graph distance defined,

$$d_G(\text{class node i, class node j}) \triangleq \text{shortest path length between nodes}$$

Details about hyper-parameters selection is described in Section 4.1.

## 3.2 Super-group retrieval algorithm

In this section we present a super-group retrieval algorithm. Returning a super-group can be useful in hard scenarios for example, when given object samples with high confusion between similar classes or when making an inference for novel classes. Soft-NLL has a better semantic generalization ability as we show below in Section 5.1. Its top-k predictions are more accurate with respect to the true class. Therefore, they can be used as a coarse-grained classifier and retrieve a better super group, compared to standard models.

Given the model top-k predictions the algorithm follows these steps:

1. **Clean the top-k predictions from less relevant predictions:**
   The model top-k predictions may be noisy. Based on the graph taxonomy we can generate a cleaner subset of the top-k predictions.

   More specifically, given the top-k predictions, we calculate a subset of them with each ith class l-hCorrectSet as follows,

   $$S_i(l, k) = \{\text{top-k predictions}\} \cap \{\text{l-hCorrectSet}_i\}$$

   Where for a given class its l-hCorrectSet is the $l$ nearest set of classes gathered from the graph taxonomy as defined by Frome et al. (2013). We choose the highest matching set, $S_C(l, k)$ where $C = \underset{i}{\operatorname{argmax}} |S_i(l, k)|$.

2. **Generate super-group candidates:**
   The set $\{A_S\}$ equals to super-group nodes which are ancestors for each class in $S_C(l, k)$.

3. **Choose the most specific super-group:**
   The most specific super group is the $a \in \{A_S\}$ which is the lowest common ancestor (LCA) of $S_C(l, k)$ generated by $LCA = \underset{a \in \{A_S\}}{\min} \sum_{s \in S_C(l,k)} d_G(a, s)$ [2].

In Appendix 7.1 we illustrate the algorithm concepts with a simple example. In Section 4.2 we discuss ways of choosing the algorithm's hyper-parameters $k$ and $l$.

---

[2]The taxonomy is a graph i.e. there may be more than one route connecting two nodes, therefore generating LCA in the mentioned way is better than by taking the deepest ancestor, $LCA = \underset{a \in \{A_S\}}{\max} \sum d_G(a, root)$.

# 4 EXPERIMENTS

The objective of this work is to develop a method for generating a semantic vision model i.e. a model which makes semantically relevant predictions even when it makes errors. Moreover, we show its advantage in super group generation in two different scenarios of selective classification and zero shot learning.

**DATASET**   ImageNet is the largest publicly available labeled image dataset, encompassing more than 14 million images that belong to more than 21K object categories (Deng et al., 2009). The object categories are nouns in the WordNet database of the English language (Miller, 1995) . A fundamental property of WordNet is its semantic hierarchical organization of concepts.

In our experiments we adopt a subset of ImageNet the ILSVRC12 dataset, which gather 1K classes that are randomly selected according to certain criteria that aim to reduce ambiguity. These 1000 classes are mutually exclusive leaf nodes of WordNet that has 820 internal nodes. It consists of approximately 1.3 million training images and 50k validation labeled images from each category. As illustrated in Figure 2, ILSVRC12 taxonomy is a complex directed graph, there may be multiple routes from the root to the leaf nodes. Its minimal spanning tree which contains the leaf nodes is highly not balanced, where routes depth range is between 6 and 18.

**METRICS**   We use several metrics in the evaluation process, each was averaged on test images. The *flat hit@k* is a standard metric 0/1 error used in large scale classification problems which returns a success if the true label resides in the top k predictions. To measure the semantic quality of predictions beyond the true label, we also evaluate with the hierarchical-precision@k (*hp@k*) introduced in Frome et al. (2013), which is a semantic relevance of the model top-k predictions. It is computed as a fraction of the top-k predictions that overlap with the true class *k-hCorrectSet*:

$$ hp@k = \frac{1}{N} \sum_{n=1}^{N} \frac{\text{number of model's top-k predictions in k-hCorrectSet for image i}}{k} $$

where for a true class k-hCorrectSet is the k nearest classes set gathered from the graph taxonomy as described in detail in Frome et al. (2013).

In order to evaluate super-group (SG) generation succession we adopt two metrics, $SG\text{-}hit$ is a 0/1 error metric, where given a predicated label, it returns success if there is a path between the SG candidate and the true class. The latter metric may be useless because the algorithm may return the root, therefore we calculate how much a candidate is specific by measuring its distance to the true class i.e. $SG\text{-}specificity = d_G(SG, truelabel)$. The last two metrics are based on the graph taxonomy. We check the SG generation succession in two scenarios: selective classification and zero shot learning.

**TRAINING DETAILS**   In our experiments we are comparing semantic ability of different loss paradigms while using the same core vision model. We train Resnet50 proposed by He et al. (2015) and Alexnet presented in Krizhevsky et al. (2012) covnvolutional networks as our core vision models. We use the data preprocessing, training procedures, and hyperparameters as described in these papers.

## 4.1 SOFT HIERARCHICAL BASED PROBABILITIES: CHOOSING HYPER-PARAMETERS

In this section we deal with determination of soft labels hierarchical probabilities hyper-parameters $\Pi$ which defined in Equation 1. Given $\Pi$ ith row which indicates the true class record. The probabilities of all jth corresponding classes $\Pi_{i,j}$ satisfy

$$ \Pi_{i,i}/\Pi_{i,j} = f_d, \ \forall \ i,j \ \text{s.t} \ d_G(i,j) = d \tag{2} $$

where $\Pi_{i,i}$ is the true class probability. That is, the probabilities of all jth classes which share a equal distance $d$ to the ith true class, are degrade by a constant *shrink factor* $f_d$ in relation to the true ith class probability. $f_d$ values are in direct relation to the on-graph distance metric. In Appendix 7.1 we give a specific example of shrink vector f and class instance.

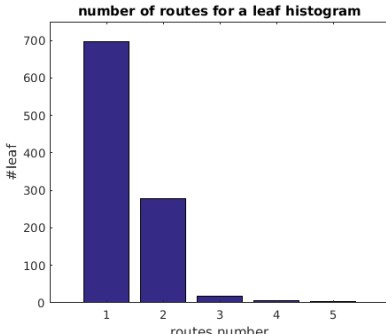
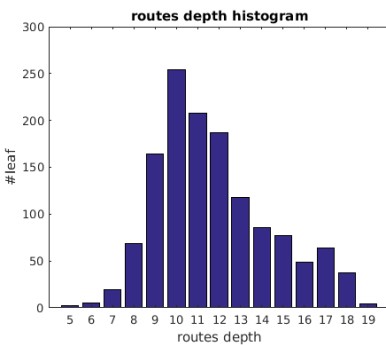

Figure 2: taxonomy statistics: number of different routes from root to leaf histogram (left), leaf routes depth histogram, remark: multiple routes per leaf included (right).

We empirically investigate the impact of different sets of $f$ values on the flat-hit and hierarchical-precision metrics on the ILSVRC 2012 1K benchmark. The pattern of the shrink vectors get the form $[f_2, f_{3,4}, f_{5,6}]$ where $f_{3,4}$ indicates that both distance level 3 and level 4 gets the same value. Distance levels that are not indicated get zero weight. To provide a baseline for comparison, we compared the performance of our model to a hard-NLL model, where all models trained with Renset50. Full results report can be found in Appendix 7.2 where Figure 3 (left) visualizes those results. Comparing 1-3 models to the baseline 0 shows that adding weight probabilities to far classes increases the semantic metric performance hp@k for growing k values. In other words, even labels which share common predecessor have valuable semantic information in different what was defined in Deng et al. (2014) which defined mutual subsumption only between labels share an immediate parent. Secondly, if we skip distance levels the performance is degraded as can be showed in model 4, where we skip all classes with distance level d=2 giving these classes a zero probability. Furthermore, using small weights yields a model which has only a minor decrease in flat-hit while still boosting the hp@k as can be seen in model 5 relative to model 3. An evidence stems from model error analysis gives validity to this mechanism in Appendix 7.3.

## 4.2 SUPER-GROUP RETRIEVAL: CHOOSING HYPER-PARAMETERS

As we mentioned an interesting and useful cases of super-group generation is for hard scenarios where the model is probable to miss its top-1 or when making an inference for novel classes. In this section we focus on the first case. According to this we take into consideration only the samples where the standard softmax-NLL makes an error in it's top1 prediction. In Section 5.2 we discuss of how to predict the failure cases and formalize a selection mechanism of returning a specific prediction when we have enough confidence and otherwise return the super-category.

The algorithm for super-group generation incorporates two hyper-parameters $k$ and $l$, i.e the number $k$ of the top-k predictions and the minimal extent of the hCorrectSet. Clearly, increasing both $k$ and $l$ improves SG-hit while hurts SG-Specificity. In this section we investigate the impact of combinations of these parameters on super-group performance. The experiments were performed with the same trained soft-NLL model used for results in Section 5.1. As baseline we compared the performance of our soft model to a standard softmax model. Both models were trained with resnet50 topology.

Our first observation is that, $k$ should not be fixed as a uniform value for all cases. It should be determined adaptively according to the uncertainty of the model top-k predictions. This uncertainty can be measured by the model *top-k probability coverage* defined,

$$\pi_\theta = \sum_{i \in top\text{-}k} \hat{\pi}_i$$

where $\hat{\pi}_i$ is the model softmax response for the ith class, and the subscript $\theta$ indicates that $\pi_\theta$ is a threshold . That is, if only small $k$ say $k = 5$ gives high probability coverage say $\pi_\theta = 0.95$ then the top-5 predictions may be considered with high certainty relative to the case when we need $k = 50$ to get the same probability coverage. This adaptive approach is needed because if we have high

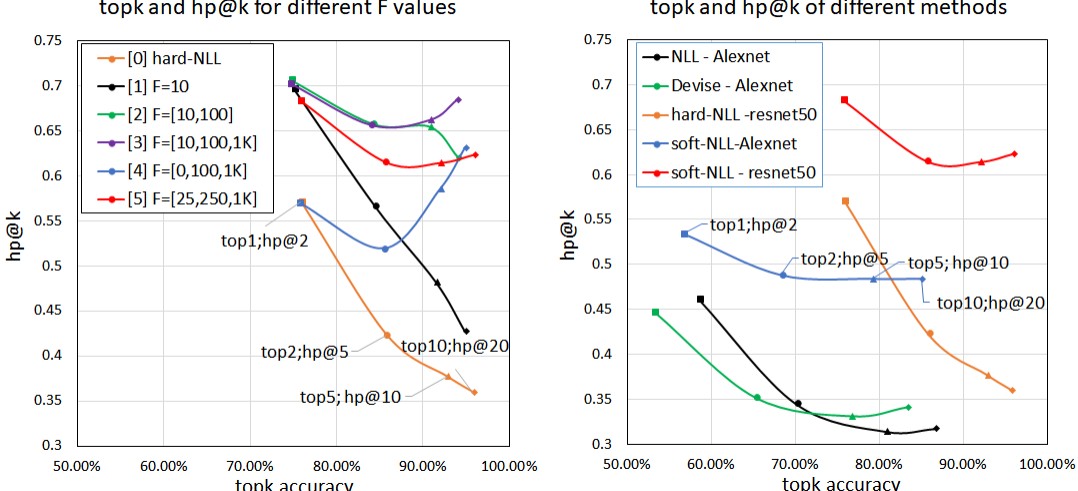

Figure 3: Visualization of flat-hit@k and hp@k metrics calculated on ILSVRC12 1K validation dataset. (Left) Performance for different $f_d$ values trained model relative to baseline hard-NLL. (Right) Comparing different loss paradigms, where soft-NLL resnet50 is the same model like 5 in left figure. Each specific set of flat-hit and hp@k point is indicated by a different symbol. The sets are displayed near it's corresponding symbol for one curve in each figure and are equivalent to all other curves. We would like to get curves which are right and up i.e. with better topk accuracy and better hp@k

confidence for small $k$ but nonetheless we choose to increase it more, then $SG\text{-}Specificity$ may be degraded. On the other hand, if we decrease $k$ in cases when $\pi_\theta$ is small i.e. low confidence we may miss the super-group. Thus, in the experiments we choose to take $k \geq 5$ while demanding that $\pi_\theta > \pi_{tresh}$, where $\pi_{thresh}$ is a fixed threshold.

Figure 4 displays both metrics $SG\text{-}Specificity$ vs $SG\text{-}hit$ for different values of k-probability coverage and $l$ values for both hard and soft models as described in the Figure caption. For both models we show that we can get super-category predictions with more than $80\%$ hit-rate when the standard model fails in returning the specific category. Moreover, these curves can give a systematic way of choosing the best $k$ and $l$. Given a demand on one metric dictates the sets of parameter to get the best in the other metric. That is for $SG\text{-}hit = 0.7$ the best lowest $SG\text{-}Specificity$ is given for soft-NLL with $l = 20$ and for $SG\text{-}Specificity = 3$ the highest $SG\text{-}hit$ is given for soft-NLL with $l = 20$. Furthermore, Soft-NLL outperformed the standard softmax model with better hit and specificity for each such demand. A handful of selected examples from ILSVRC12 validation dataset shows a qualitatively illustration of the benefits of super-group generation by using our proposed algorithm and soft-model in Appendix 7.4.

## 5 RESULTS

### 5.1 SEMANTIC RELEVANCE

This section presents flat and hierarchical results on the ILSVRC 2012 1K dataset. We compare our soft-NLL to standard softmax-NLL using two architectures Resnet50 and Alexnet, and to DeVise approach presented in Frome et al. (2013) trained with Alexnet. The hyper-parameter search setup used with soft-NLL loss is specified in details in Section 4.1

Figure 3 (right) visualizes these results. The soft-NLL shows a significant improvement in its semantic generalization ability while exhibiting only a minor decrease in the top-k accuracy compared to standard softmax-NLL on a same deep topologies. For Resnet50 at k=5 the soft-NLL gives 0.615 while hard-NLL gives 0.423 model which is about 45% relative improvement, at k=10 soft-NLL and hard-NLL gives 0.614 and 0.377 respectively which is a 63% relative improvement. Our rel-

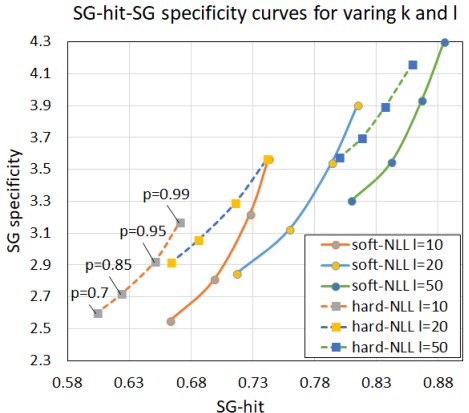

Figure 4: SG-hit and SG-specificity metrics calculated on top-1 miss cases of hard-NLL model in ILSVRC12 1K validation dataset. The effect of varying k and l on SG generation for soft-NLL and hard-NLL models trained with resnet50. For each $l$ we use a set of k-probability coverage: 0.7,0.85,0.95,0.99 which are arranged from left to right for each curve. In order to do well We would like to get curves which are right and bottom i.e. with better hit and more specific

ative improvement is in order of magnitude compared with the improvement given by DeVise, for k=5 and k=10 DeVise Frome et al. (2013) method gives a relative improvement of about 2% and 5% respectively over the standard model. A handful of selected examples from ILSVRC12 validation dataset shows a qualitatively illustration of our model in Appendix 7.4.

## 5.2 HIERARCHICAL SELECTIVE CLASSIFICATION

When we consider a large-scale problem ($|Y| \gg 1$), the standard approach is to consider the top-k predictions where $k > 1$. But in practice, given an image a desired need may be to get a single label and not k noisy predictions. However, considering only the top-1 prediction includes many mistakes even in state-of-the-art models. As we discuss in Section 4.2 Soft-NLL can be used to make higher quality reasonable super-category inferences given hard samples because of it has a better semantic generalization ability as shown before in Section 5.1. In this section we discuss about a selection mechanism of returning only a single high probable prediction. That is return the top-1 specific prediction when we have enough confidence and otherwise returning the super-class using owe preferred model. In other words, we formalize this mechanism and discuss of how to identify if the top-1 prediction is improbable to make hit.

A selection with guaranteed risk control problem is formalized in Geifman & El-Yaniv (2017), where given a classifier $f$ a *selective-classifier* returns $f(x)$ prediction if it guaranteed a desired error and otherwise it abstains the prediction. In our case instead of abstaining the predication if it has low confidence we propose to retrieve the super-group as follows,

$$(f,g) \triangleq \begin{cases} top\text{-}1(f_{NLL}(x)), & \text{if } g(x) = 1 \\ SG(top\text{-}k(f_{soft-NLL}(x))), & \text{if } g(x) = 0 \end{cases}$$

where, $f_{NLL}(x)$,$f_{soft-NLL}(x)$ are the standard and semantic classifiers and $g : X \rightarrow \{0, 1\}$ is a selection function which calculated according to the desired risk.

As showed in Geifman & El-Yaniv (2017) the top-1 softmax model response can be used to control the guaranteed error that is,

$$g(x) = \begin{cases} 1, & \text{if } SR_{top\text{-}1}(x) > \theta \\ 0, & \text{otherwise} \end{cases}$$

where $\theta$ is a threshold determined according the desired error of top-1 predictions. That is, suppose the model gives a total 75% top-1 accuracy while we would like a desired error of 5% (or 95% top-1 accuracy) this demand impose $\theta_o$ to guarantee this risk. In other words, we coverage only a subset of our data distribution such that we fulfill satisfactory demands.

Given a classifier a risk-coverage curve presents the computed risk for many $\theta$ values. We calculated risk-coverage curve for ILSVRC12 validation dataset. For each $\theta$ value we split the set. In the guaranteed-controlled set we calculated top-1 error and in the other complementary split we calculated the super group generation ability. We inverse the x-axes of the latter split set for alignment reasons. As shown in 5. For example, for 65% coverage NLL gives about 92% top-1 accuracy, and

in the complementary set (35% coverage) we get $SG\text{-}hit$ of 87% and 3.45 $SG\text{-}specificity$ using soft-NLL model, while without this mechanism we can obtain only 75% total accuracy on the full coverage set. Those super-categoty inferences can then be used with fined-grained classifier to obtain a more specific category. Moreover, our soft-NLL model outperforms standard NLL in both SG metrics for all $\theta$-coverage values. This results show that our semantic soft model returns higher quality super-group classes.

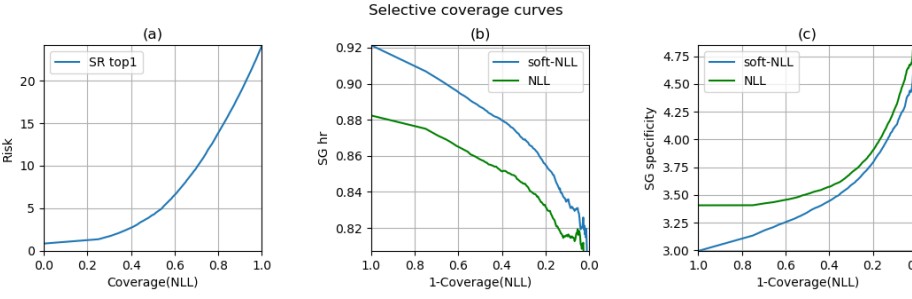

Figure 5: Selective coverage curves, (a) top-1 risk using standard NLL softmax, (b-c) super-group hr and specificity in the complementary sets. The x-axes of (b)-(c) was inverted for relation to (a) easier.

### 5.3 HIERARCHICAL ZERO SHOT LEARNING

Soft-NLL model can be generalized as a coarse-grained descriptor to return a better super group. In this manner, our model can make reasonable inferences about candidate it has never visually observed and given a novel class return its super-class, we call this inference **Hierarchical Zero shot learning**. Our definition is easier from the common one which aims to accurately recognize the unseen classes as in Frome et al. (2013); Norouzi et al. (2013); Changpinyo et al. (2016). We think that this super-category inference can be reasonable if we can get satisfactory results in our mode relative to the poor results obtained even with state of the art methods with the common definition (Xian et al., 2017; Changpinyo et al., 2017). To test this hypothesis, we extracted images from the ImageNet 2011 21K dataset with labels that were not included in the ILSVRC 2012 1K dataset on which our model was trained.

The zero-shot experiments were performed with the same trained Soft-NLL model used for results in Section 5.1. To provide a stronger baseline for comparison, we compared the performance of our model to standard softmax model. We again evaluate super group quality with SG-hit and SG-specificity metrics.

To quantify the performance of the model, we constructed from ImageNet 2011 21K zero-shot data two test data sets as defined in Frome et al. (2013). The datasets are defined with increasing difficulty based on the image labels' graph distance from ILSVRC 2012 1K labels. '2-hop' is the easy dataset which comprised of 1,548 labels that are within two tree hops of the training labels, making them visually and semantically similar to the training set. A more difficult '3-hop' dataset comprised of 5,989 labels was constructed in the same manner. The datasets contains 1.3 million and 4.6 million images respectively.

Figure 6 compared between soft-NLL and hard-NLL $SG\text{-}Specificity$, $SG\text{-}hit$ metrics for different values of k-probability coverage and $l$ values. Both models can give more than 70% SG-hit with reasonable SG-specificity even in the harder benchmark, while state of the art methods (Xian et al., 2017; Changpinyo et al., 2017) gets less then 5% on the harder set trying to hit the true label. We think that it is apparently better to give a high level inference with good correctness than trying to give very specific inference with very low confidence. Moreover, based on ours super-group predictions a more specific class can be generated using a fined-grained mechanism. Secondly, our soft-NLL outperformed hard-NLL on both metrics on both datasets. That is, it gets a better hit for each demand in the SG-specificity metric and a better specific super-class for each demand in the hit meter. Futhermore, in the harder '3-hop' dataset we get some degradation in the SG-hit metric but

get a comparable results on the SG-specificity metric relative to '2-hop' dataset. The specificity of predictions is reasonable taking into consider the high degree of classes as shown in 2.

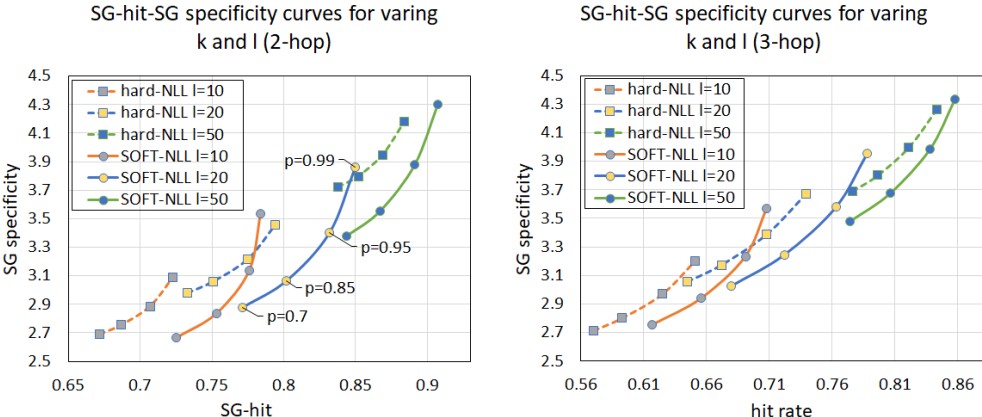

Figure 6: SG-hit and SG-specificity metrics calculated on zero-shot '2-hop'(left) and '3-hop' ima-genet11 datasets. Comparision of hard-NLL and soft-NLL performance for varing k and l, where for each $l$ we use a set of k-probability coverage: 0.7,0.85,0.95,0.99 which are arranged from left to right for each curve. In order to do well We would like to get curves which are right and bottom i.e. with better hit and more specific

## 6 CONCLUSION

In this work we have shown that our hierarchical probability based soft-NLL loss can be trained to give a comparable performance to the state-of-the-art hard-NLL softmax based model on flat classification metric, while simultaneously making more semantically reasonable errors, which is indicated by its significant improved performance on a hierarchical label metric relative to existing methods (Frome et al., 2013). Our model's learning process has no additional cost in training and classification time. This approach is independent of the specific deep architecture and thus can always benefit from design progress.

Furthermore, we present an algorithm where given a probabilistic model, based on its top-k predictions our algorithm returns the input corresponding super-group based on classes hierarchy without any further learning. Using this technique we can take into account only a single high probable prediction instead of taking top-k predictions for multi-class large-scale problems. We have shown that our soft-NLL retrieve a higher-quality coarse category on two use cases than the standard model. The first case is when the regular model is probable to make a miss. We show that our algorithm obtains more than 80% hit-rate when the standard model fails in returning the specific category. The second is for zero-shot scenario which deals with making an inference for novel classes. In this case our soft-model success in returning a high-quality with success of more then 75% while the state of the art methods gets less than 5% trying to return a specific super-class. Based on those super-group predictions a more specific class can be generated using a fined-grained mechanism.

Moving forward, we are experimenting cases where the taxonomy is unknown. Bilal et al. (2018) proposed method to constructing the class hierarchy based on identifying block pattern in the confusion matrix, then refining the hierarchy recursively. Such taxonomy is not perfect and probably contains noises. We will deal how these significantly affect the models performance. Moreover, We are experimenting an adaptive use of our loss function according to visual similarity between classes resides in the confusion matrix. In some cases a hierarchical closeness may not coincide with visual closeness which is the integral issue when trying to differentiate between visual objects. Another interesting work we are going to to work on is whether our semantic inferences are less easily fooled relative to standard models (Nguyen et al., 2015) and thus can help to get higher confidence networks.

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

## 7 APPENDIX

### 7.1 APPENDIX A: ADDITIONAL EXAMPLES

**Super-group retrival algorithm example:** Suppose we have a simple taxonomy as illustrated in Figure 7, where the leaf nodes are valid classes and the other nodes are theirs super-classes. Suppose we got a sample from class 1, and our top-3 predictions are: 0,1 and 6. Each class has its own 3-hCorrestSet, where the true class hCorrestSet gives the maximal intersection set with the top3 predication i.e. $S_1(3,3) = 0, 1$. We have three super-group candidates: A,B,D where D is the LCA generated super-group.

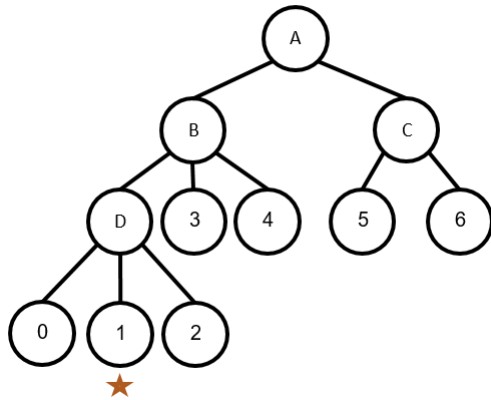

Figure 7: Super-class generation example

**Soft hierarchical based probabilities example:** Suppose the shrink vector is $[f_2, f_{3,4}, f_{5,6}] = [10, 100, 1000]$, where $f_{3,4}$ indicates that both distance level 3 and level 4 gets the same value. Table 1 describes the probability values given for the class 'tabby',

| distance level $[d]$ | #classes | shrink factor $[f_d]$ | Probability weights |
|---|---|---|---|
| 0 | 1 | 1 | 61.9% |
| 2 | 4 | 10 | 6.19% |
| 3 | 0 | - | - |
| 4 | 9 | 100 | 0.619% |
| 5 | 29 | 1K | 0.0619% |
| 6 | 94 | 1K | 0.0619% |
| $\geq 7$ | 863 | $\infty$ | 0 |

Table 1: Soft hierarchical probabilities for a specific class ('tabby') with the shrink vector $[f_2, f_{3,4}, f_{5,6}] = [10, 100, 1K]$

### 7.2 APPENDIX B: ADDITIONAL TABLE RESULTS

**Soft hierarchical based probabilities - choosing hyper-parameters results:** Table 2 indicates the impact of different sets of $f$ values on the flat-hit and hierarchical-precision metrics on the ILSVRC 2012 1K benchmark. To provide a baseline for comparison, we indicate the performance of our model next to hard-NLL model, where all models trained with Renset50.

**Semantic relevant results:** Table 3 display flat and hierarchical metrics for all experimental setting outlined in Section 5.1

| Model name | Flat hit@k (%) | | | | Hierarchical precision@k | | | |
|---|---|---|---|---|---|---|---|---|
| | 1 | 2 | 5 | 10 | 2 | 5 | 10 | 20 |
| [0] hard-NLL | 75.85 | 85.79 | 92.81 | 95.75 | 0.571 | 0.423 | 0.377 | 0.360 |
| [1] $f_2 = 10$ | 75.12 | 84.47 | 91.48 | 94.85 | 0.696 | 0.566 | 0.482 | 0.427 |
| [2] $f_2, f_{3,4} = 10, 100$ | 74.74 | 84.24 | 90.81 | 94.04 | 0.706 | 0.657 | 0.654 | 0.620 |
| [3] $f_2, f_{3,4}, f_{5,6} = 10, 100, 1K$ | 74.59 | 84.04 | 90.81 | 93.91 | 0.702 | 0.655 | 0.662 | 0.684 |
| [4] $f_2, f_{3,4}, f_{5,6} = 0, 100, 1K$ | 75.64 | 85.49 | 92.01 | 94.92 | 0.570 | 0.519 | 0.585 | 0.631 |
| [5] $f_2, f_{3,4}, f_{5,6} = 25, 250, 1K$ | 75.73 | 85.63 | 92.04 | 95.95 | 0.683 | 0.615 | 0.614 | 0.623 |

Table 2: Shirk factor $f_d$ impact on model performance on ImageNet ILSVRC12 1K validation set trained with Resnet50

| Model name | Flat hit@k (%) | | | | Hierarchical precision@k | | | |
|---|---|---|---|---|---|---|---|---|
| | 1 | 2 | 5 | 10 | 2 | 5 | 10 | 20 |
| NLL-Alexnet | 58.6 | 70.1 | 80.8 | 86.7 | 0.461 | 0.345 | 0.314 | 0.317 |
| DeVise(dim=500) | 53.2 | 65.2 | 76.7 | 83.3 | 0.447 | 0.352 | 0.331 | 0.341 |
| NLL-Resnet50 | 75.9 | 85.8 | 92.8 | 95.8 | 0.571 | 0.423 | 0.377 | 0.360 |
| soft-NLL-Alexnet | 57.7 | 69.2 | 79.7 | 85.6 | 0.542 | 0.493 | 0.491 | 0.490 |
| soft-NLL-Resnet50 | 75.7 | 85.6 | 92.0 | 96.0 | 0.683 | 0.615 | 0.614 | 0.623 |

Table 3: Comparison of model performance on test set (ImageNet ILSVRC12 1K validation set)

### 7.3 APPENDIX C: EXPLANATION OF SEMANTIC WEIGHTING TECHNIQUE

We found an interesting evidence which gives validity and intuition to our semantic empirical weighting technique proposed in Section 4.1. This evidence stems from analyzing the error hard-model behavior. We found that this behavior is common to different deep architectures, where in this section the details regarded to resnet50 topology trained on ILSVRC12-1K dataset. All labels in this dataset are defined as leaf nodes in the taxonomy graph as mentioned in Section 4. Figure 8 (a) presents the Top-1 hard model error prediction occurrences at each distance level, where the sum of all columns gives the total model error. Although we expect that this graph will decrease monotonically according to the distance [3] it has much complex behavior. As expected there is a maximum in level-2 which is the closest related leaf nodes distance level, means that the the maximum confusion corresponds to taxonomy distance similarity. Moreover, starting from level-7 there is a monotonic decrease behavior. However, in level-3 there is a decrease and an immediate salient increase in level-4 which is opposed to intuition.

Nevertheless, We found that the deep model is truly semantic and acts according to our intuition. The salient increase in level-4 is a biased term stems directly from dataset distribution. As shown Figure 8 (b) given a leaf node, there is a significant relative increase in the occurrences of leaf neighbours in level-4 relative to level-3 (about 3 times). A dataset bias term can be expressed by a multiplication argument. That is, the model behaviour can be formalize as,

$$M_d = N_d \cdot f_d \tag{3}$$

where, $M_k$ is the empirical model probability of error at distance d, which is biased according to data as we just claim. This term is shown in Figure 8 (a). $N_k$ is the dataset probability occurrence at distance d and $f_d$ is the true model preferring factor of level d excluding the dataset bias term. Figure 9a presents the behaviour of f. f decreases approximately exponentially in d according: $f = 6.46 \cdot 10^{-0.135 \cdot d}$, which proves our claim.

This fact is an evident for the ability of CNN models to make a semantic errors in average, although they did not make any use in hierarchical information, because visual object categories are naturally hierarchical (Deng et al., 2010; Bilal et al., 2018). Nevertheless, using directly semantic hierarchy can improve classification systems. f can be interpreted as a similarity metric between categories at different distance level and thus it can be used as a guideline of to define $\Pi$ the similarity probability

---

[3]ILSVRC12-1K is a huge and diverse benchmark, therefore averaging on the dataset is probably to give strong unbiased claims

weighting matrix presented in Section 3.1. Each class gets the maximum weight, and the similarity weight is divided according to distance level. Because of the fast decay of f, labels which are in distance level-7 can be viewed as labels which are fully disjoint categories and therefore get zero weights. Furthermore, We tried to use f values directly to calculated Π values and got weights that are close to what we found through experiments. We even get some improvement in the hierarchical metric as shown in Figure 9b. We assume that our weighting mechanism can be applied to other cases because our mechanism was deduced from the largest publicly dataset and we further remove biased influence.

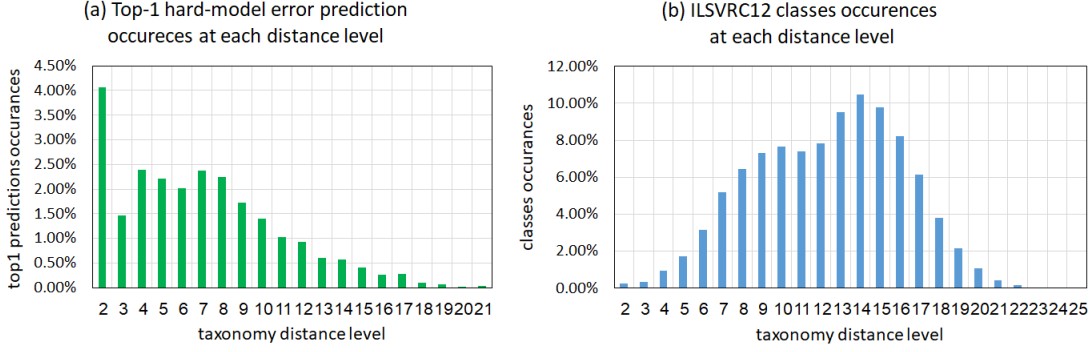

Figure 8: (a) Top-1 hard-model error prediction occurrences at each distance level. (b) ILSVRC12 classes occurrences at each distance level calculated by averaging all classes.

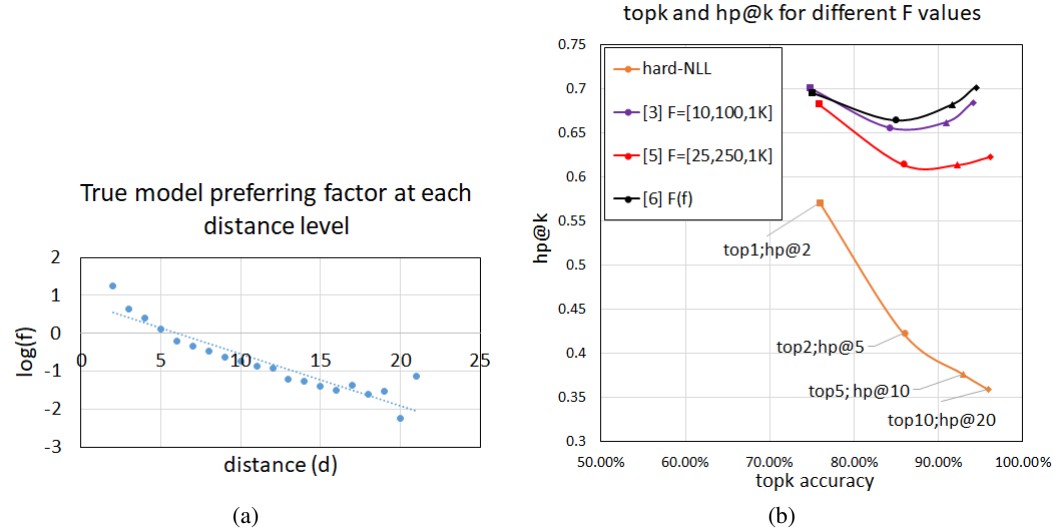

Figure 9: (a) True model preferring factor at each distance level d. (b) Visualization of flat-hit@k and hp@k metrics calculated on ILSVRC12 1K validation dataset. Comparing between between different soft weighting paradigms to hard-NLL, where all models trained with resnet50. [6] denotes the experiments where Π values calculated directly from f values, and [3,5] are the empirical results presented in Section 4.1

## 7.4 APPENDIX D: SEMANTIC HAND-NLL VS. SOFT-NLL SAMPLES

Figure 10 qualitatively illustrate our soft model. We selected examples from ILSVRC12 validation dataset and make a comparison between our soft-model to the hard-model. Both models were trained with resnet50 topology, where our model is the same trained model used for results in Section 5.1.

For each example we show the top-5 label predictions and theirs corresponded softmax responses. Next to each class we indicates taxonomy distance from true class to that class. In all cases our model predictions are generally more semantically close to the desired label. Moreover, The soft top-5 ranks are mostly arranged according to their relevance to the true class.

Figure 10 (a-c) shows cases where our model accurately predict the true class in the top-1 place where hard-NLL missed it. In Figure 10a the hard-model top-5 classes are: a kind of a fox, the true class i.e. a siamese cat and three dog breeds while our model returns different types of domestic cat. Furthermore, our model outputs are less biased to irrelevant context. It is very reasonable that the hard-model choose arctic-fox mostly because of the white background which is similar to snowy environment. Figure 10b shows a case where our model identify the true class in top-1 although the texture similarity to other top-5 classes. Figure 10c image contains two objects where both are resides in the set. Our model gives to both these labels high score where the hard-model give most of probability weight to one of the classes. Figure 10 (d-e) shows cases where the hard-model accurately predict the true class in the top-1 place where our model missed it. In Figure 10d although our model identify that the class is a camera it confuses between the two camera types. This is reasonable because through learning a specific features of a class our model influence other close related classes too. Figure 10e is an image which contains two relevant objects. Our model predict with high confidence the class 'punching bag' which is a kind of a ball in the taxonomy, because many punching bags in training images color is red like the ball in the image. Nevertheless, it still identify the true class in its top-5 predictions.

### 7.5 APPENDIX E: SUPER-GROUP RETRIEVAL IN HARD CLASSIFICATION CASES SAMPLES

Figure 11 qualitatively illustrates the significant benefit of our proposed super-group retrieval algorithm presented in Section 3.2. We selected examples from ILSVRC12 validation dataset where the hard-model miss its top-1, a,d,e are cases where it miss its top-5. We used our soft-model as the core-model of our proposed algorithm, because its improvement over the hard-model as shown in Section 4.2. We choose $l - hCorrectSet$ extent to be 10 and adaptive k such that $k \geq 5$ while demanding that $\pi_\theta > 0.85$ with those hyper-parameters we can get $SG\text{-}hit = 0.7$ and $SG\text{-}Specificity = 3$ as shown in Figure 4. Both hard and soft models were trained with resnet50 topology, where our model is the same trained model used for results in Section 5.1. For each example we show the hard-model top-5 label predictions and theirs corresponded softmax responses. Next to each class we indicates taxonomy distance from true class to that class. Moreover we indicates in the table the true class next to the soft-SG prediction label, SG hierarchical distance to the true class and the number of soft predictions used in order to generate the SG.

All samples are cases where it is not trivial to guess what is the object given the top-5 predictions. For example in Figure 11a it may be a snake or a lizard, while our method succeed in identify even the specific breed of the snake. In Figures d-e the confidence is very low such that we scale the limits of the responses accordingly. In those cases we cannot get even a clue about the inner object based on the hard-model predictions. Our soft-model used k=50 predictions and succeed in predicting a very reasonable super-class outputs.

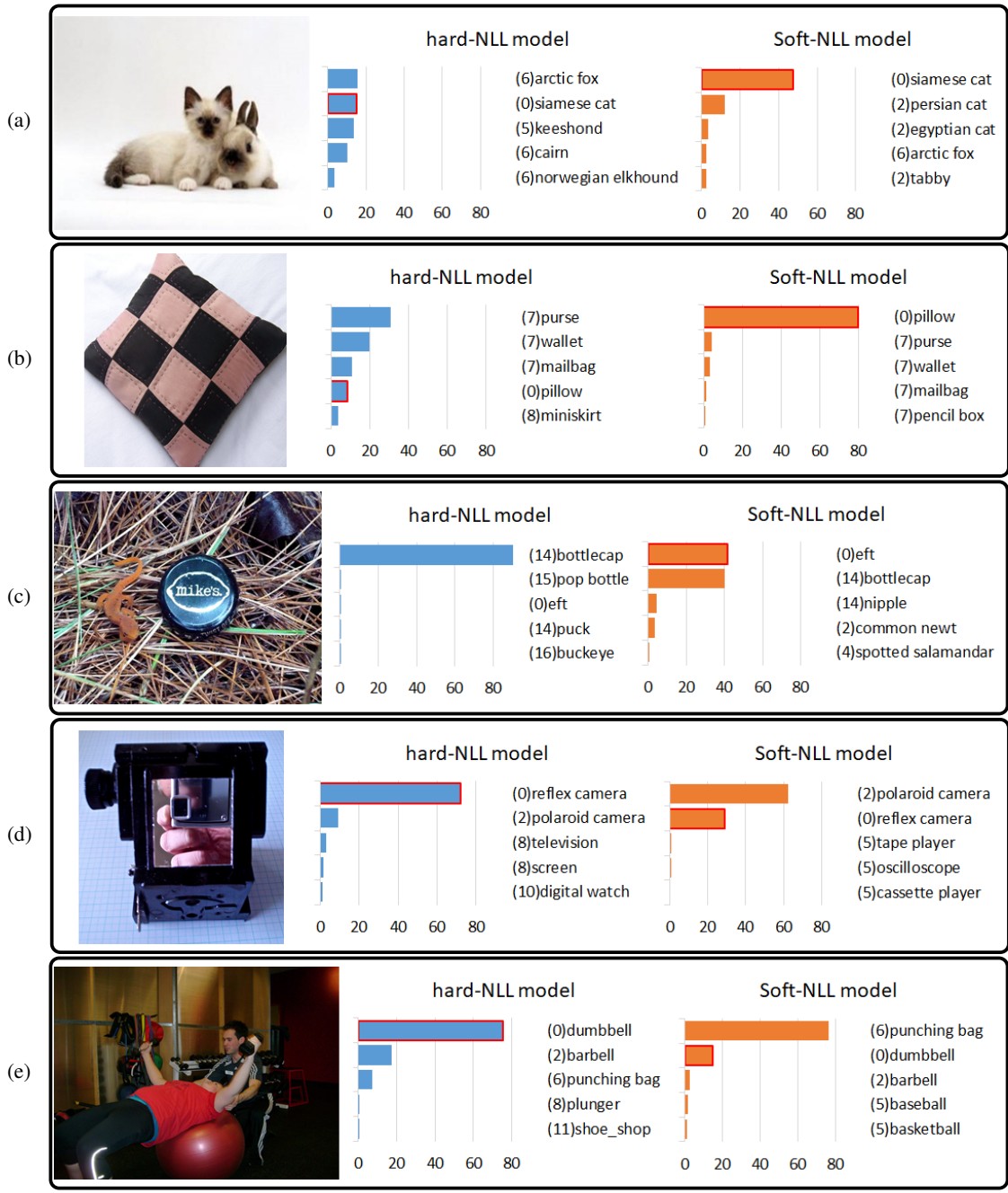

Figure 10: Comparison examples of hard-NLL and soft-NLL top-5 predictions softmax responses (%) taken from ILSVRC2012 validation set. The responses are arranged from high to low values. The true class is indicated by the corresponded column red border. The number next to class name indicates WordNet taxonomy distance from the true class calculated to that class

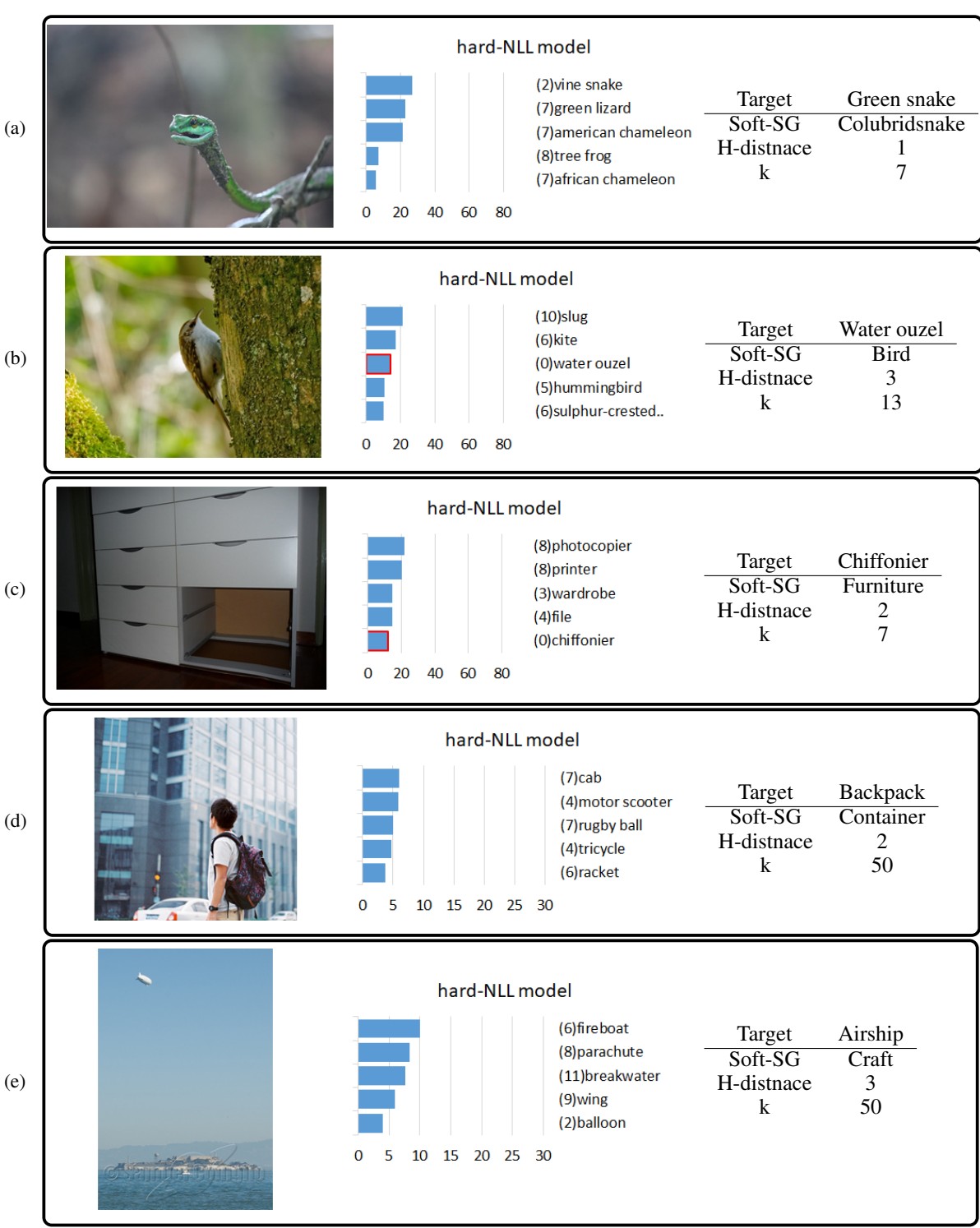

Figure 11: soft super-group retrieval examples. All examples are cases where hard-NLL miss its top-1 prediction, taken from ILSVRC12 validation set. For each examples we show hard-NLL top-5 softmax responses (%), which are arranged from high to low values. The number next to class name indicates WordNet taxonomy distance from the true class calculated to that class. Moreover, we indicates the true class, the soft SG prediction name, SG hierarchical distance from the true class and the number of soft predictions used in order to generate the SG (k)

