# OpenReview forum: "DEEP HIERARCHICAL MODEL FOR HIERARCHICAL SELECTIVE CLASSIFICATION AND ZERO SHOT LEARNING"
_ICLR.cc/2019/Conference_

### Official Review · AnonReviewer2 · 2018-10-30
**The paper violates the double blind review policy**

**Rating:** 2
**Confidence:** 4

**Review:**

First of all, the paper cannot be accepted because it violates the double blind submission policy by including an acknowledgments section.

Nonetheless, I will give some brief comments:

 The paper proposes a probabilistic hierarchical approach to perform zero-shot learning.
Instead of directly optimizing the standard cross-entropy loss, the paper considers some soft probability scores that consider some class graph taxonomy.

 The experimental section of the paper is strong enough although more baselines could have been tested. The paper only compares the usual cross entropy loss with their proposed soft-classification framework.
Nonetheless, different architectures of neural networks are tested on ImageNet and validate the fact that the soft probability strategy improves performance on the zero-shot learning task.


On the other hand, the theoretical aspect is weak. The proposed method seems to be a straightforward extension of Frome et al., NIPS 2013. The main contribution is that soft probability scores are used to perform classification instead of using only class membership information.

Some weighting strategy is proposed in Section 2.2 but the proposed steps seem very ad hoc with no theoretical justification. The first equation on page 8 has the same problem where some random definition is provided.

---

> ### Author Response · Authors · 2018-11-06
> **response to reviewer 3**
>
> hi,
> First, thanks on reviewing my work. your feedback is very important to me.
> 1) This is my first published academic doc, so I worked according to the guidelines that I saw. I didn't see a direct guideline about the ack part. So made this mistake because lacking experience. I can propose that for next a direct guideline for blind names from ack. will be mentioned as it done in the authors names part.
> hope that u can reply to me although about next issues….
>
> 2) The article has two main novelties the first is as u mentioned. The second is the proposed algorithm where given a probabilistic model it can return 'good' super-class based only on the taxonomy without another learning process.  I showed that the soft model gives better super-class on two scenarios.
> 3) You mentioned that the experiment part is weak. Which experiments should I add?
>  -  In order to prove the semantic generalization ability, I compared the soft-model to hard-model using two different topologies (Alexnet and Resnet50) and to DeVise.
>  - In order to prove that the model has better super-class. I compared the soft-model to the hard-model  on the two scenarios.
> 4) Can u please give me a reference to an article about zero-shot where this soft-taxonomy based is made?
> 5) Truly I used Frome proposed metric to measure the semantic ability of a model. But, our work is completely different. Frome is an embedding based solution. I propose a loss function which exploits the inter-class hierarchy.
> 6) I justify the weighting through experiments which visualizing in Fig. 2 left and table 2.
> I can add another theoretical justification in the appendix if u give me another chance.
> 7) About the equation in page 8, I put a reference where I found a version of this equation.
> the work of defining the selective term is not mine.
> Best Regards, Thx

---

### Official Review · AnonReviewer3 · 2018-10-31
**An interesting paper but can still be improved.**

**Rating:** 5
**Confidence:** 3

**Review:**

This paper proposes a new soft negative log-likelihood loss formulation for multi-class classification problems. The new loss is built upon the taxonomy graph of labels, which is provided as external knowledge, and this loss provides better semantic generalization ability compared to a regular N-way classifier and yields more accurate and meaningful super-class predictions.

This paper is well-written. The main ideas and claims are clearly expressed. The main benefits of the new loss are caused by the extra information contained by the taxonomy of labels, and this idea is well-known and popular in the literature. Based on this reason, I think the main contribution of this paper is the discussion on two novel learning settings, which related to the super-classes. However, the formulation of the new soft NLL loss and the SG measurement involves lots of concepts designed based on experiences, so it’s hard to say whether these are the optimal choices. So, I suggest the authors discuss more on these designs.
Another thing I concern about is the source of label taxonomy. How to efficiently generate the taxonomy? What if the taxonomy is not perfect and contains noises? Will these significantly affect the models’ performance? I think it’s better to take these problems into consideration.
In conclusion, I think this is an interesting paper but can still be improved.

---

> ### Author Response · Authors · 2018-11-27
> **response to reviewer 2**
>
> Hi,
> Thanks on your feedback.
> Truly, as u wrote the super-group concept is truly very interesting and novel.
> I emphasis this in the zero-shot results part and in conclusion sections. Although the super-group concept is based on understanding the data and benefits from the soft-model, the result are fantastic. and need to be considered to be publish in the conference.
> A) for zero-shot learning state of the art methods gives about 5% top1 accuracy,  while trying to give a specific class. While our method gives more than 70% with identifying a valid super-category which is close related to the true class.  Once you have a super-class one can used fined-grained method to give even more specific class.
> B) Another interesting result is when trying to retrieve super-class in the cases where the standard model miss the top-1. We obtain in this case more than 80%. You can look on those samples which are very impressive.
>
> I think that the concept of the soft-weighting mechanism is truly straight-forward, but it is very interesting too. In order to give more insight on this part I add an explanation and justification in appendix C.
>
> Truly, the taxonomy labeling issue can be an interesting investigation. I tried to make some experiments in this direction, but I still have more work to do.  At this stage I added this issue to future work in conclusion part.
> Moreover, I indirectly refer to this issue in the part SUPER-GROUP RETRIEVAL: CHOOSING HYPER-PARAMETERS.
> I tried to see what is the impact of different f values, which compared to hard-model. In those cases I tried to show what is the impact of weighting far and more far distance levels.  another case is where I skip a full distance level. From those cases we can get an intuition about absence of information in the taxonomy.
>
> Beyond those issues, I made much editing work,  put more related work,  and illustrate the soft-model benefits with many examples.
>
> Best.

---

### Official Review · AnonReviewer1 · 2018-11-03
**Missing key references**

**Rating:** 4
**Confidence:** 4

**Review:**

SUMMARY
The paper presents a method for classification which takes into account the semantic hierarchy of output labels, rather than treating them as independent categories. In a typical classification setup, the loss penalizes the KL-divergence between the model’s predicted label distribution and a one-hot distribution placing all probability mass on the single ground-truth label for each example. The proposed method instead constructs a target distribution which places probability mass not only on leaf category nodes but also on their neighbors in a known semantic hierarchy of labels, then penalizes the KL-divergence between a model’s predicted distribution and this target distribution. This model is used for classification on ImageNet-1k, and for zero-shot classification on ImageNet-21k where a model must predict superclasses seen during training for images of leaf categories not seen during training.

Pros:
- Method is fairly straightforward
- Modeling relationships between labels is an important problem

Cons:
- Missing references to key prior work in this space
- Minimal comparison to prior work
- Confusing experimental setup
- Paper is difficult to read

MISSING REFERENCES
This paper is far from the first to consider the use of a semantic hierarchy to improve classification systems; see for example:

Deng et al, “Hedging your bets: Optimizing accuracy-specificity trade-offs in large scale visual recognition”, CVPR 2012

Deng et al, “Large-scale object classification using label relation graphs”, ECCV 2014 (Best Paper)

Jiang et al, “Exploiting feature and class relationships in video categorization with regularized deep neural networks”, TPAMI 2017

None of these are cited in the submission. [Deng et al, 2014] is particularly relevant, as it considers not just “is-a” relationships as in this submission, but also mutual exclusion relationships between categories. Without citation, discussion, and comparison with some of these key pieces of prior work, the current submission is incomplete.

COMPARISON TO PRIOR WORK
The only direct comparison to prior work in the paper is the comparison to DeViSE on ILSVRC12 classification performance in Table 3. However since DeViSE was intended to be used for zero-shot learning and not traditional supervised classification, this comparison seems unfair.

Instead the authors should compare their method against DeViSE and ConSE for zero-shot learning. Indeed, in Section 4.3 the authors construct a test set “in a [sic] same manner defined in Frome et al” but do not actually compare against this prior work.

I suspect that the authors chose not to perform this comparison since unlike DeViSE and ConSE their method cannot predict category labels not seen during training; instead it is constrained to predicting a known supercategory when presented with an image of a novel leaf category. As such, the proposed method is not really “zero-shot” in the sense of DeViSE and ConSE.

EXPERIMENTAL SETUP
From Section 3.1, “we adopt a subset of ImageNet the ILSVRC12 dataset which gather [sic] 1K classes [...]”. The 1000 category labels in ILSVRC12 are mutually exclusive leaf nodes; when placed in the context of the WordNet hierarchy there are 820 internal nodes between these leaves and the WordNet root. As a result, for the method to make sense I assume that all models must be trained to output classification scores for all 1820 categories rather than the 1K leaf categories. This should be made more explicit in the paper, as it means that none of the performance metrics reported in the paper are comparable to other results on ILSVRC12 which only measure performance on the 1K leaf categories.

The experiments on zero-shot learning are also confusing. Rather than following the existing experimental protocol for evaluating zero-shot learning from [Frome et al, 2013] and [Norouzi et al, 2013] the authors evaluate zero-shot learning by plotting SG-hit vs SG-specificity; while these are reasonable metrics, they make it difficult to compare with prior work.

POOR WRITING
The paper is difficult to follow, with confusing notation and many spelling and grammatical errors.

OVERALL
On the whole, the paper addresses an important problem and presents a reasonable method. However due to the omission of key references and incomplete comparison to prior work, the paper is not suitable for publication in its current form.

---

> ### Author Response · Authors · 2018-11-28
> **reply to reviwer 1**
>
> Hi,
> Thanks you very much on your detailed feedback. I will reply according to the issue you mentioned.
>
> REFERENCES PART:
> You are truly right in that.  One of my colleague told me too that I need to improve this issue, and I had worked on this issue mainly before I got your feedback.  In my last revision I improved this issue even more. I find interesting claims regards to “Large-scale object classification using label relation graphs” work. In my results I show that their exclusion mechanism may be too strict, by giving similarity
> weight even to labels which share predecessor we improve semantic ability.
>
> COMPARISON TO PRIOR WORK:
> 1) I tried to work on this issue too. I go over many articles, but I find only few works which report a their  semantic metric like hp@k, unfortunately all those works report their performance only on small datasets like: cifar100 or AWA, which are relatively to ILSVRC12. Our aim is to deals with semantics in large-scale scenarios, which are far more comlpex. Because of this issue I didn't refer to those work.
> I compared my work to the standard hard-model and to DeVise.  DeVise has two parts in their work and one of it's claims is that this model has a good semantic ability.  In the first experiment I refer to this claim.
> I added samples of my soft-model results which shows many interesting semantic abilities aspects.
> 2) Regards the zero-shot issue. In the new revision I emphasis that we are dealing with a soften version of zero-shot which I called hierarchical zero-shot. Nevertheless, I think that our method is very reasonable,  while state of the art methods on zero-shot learning which try to give a specific class, gives about 5%. Our method gives more than 70% with identifying a high quality super-category i.e  which is close related to the true class. Once you have a super-class one can used fined-grained method to give even more specific class.
> 3) Another interesting result regarded the benefits in super-group retrieval is where the standard model miss the top-1. We obtain on this case more than 80%. You can look on those samples.
>
> EXPERIMENTAL SETUP.
> 1) Sorry, but I think that maybe I wasn't very clear in my first revision. The standard and our soft-models trained only on the leaf nodes. That is these classifiers return scores for ILSVRC12-1K classes only.
> I didn't make any learning on the ancestors of the 1K classes.
> 2) I proposed an algorithm which gets a probabilistic model as mentioned.
> Moreover, by assuming that the leaf's taxonomy is known our algorithm can used this taxonomy to return super-class. That is I can tell who are the ancestor from knowing who are the relevant children.
> 3) Super-group retrieval as I mentioned is a novel concept. Therefore, I define a new evaluation metric and couldn't compare to prior work directly. The two cases deals with scenarios when trying to return the specific category we get very poor results I indicate this, In such cases we can benefit from super-class if we get it with significant performance improvement. As I shown.
> I added my results statistics and added more illustrations which show the advantages of this issue.
>
> WRITING
> I made much work with it. I added samples of models outcomes. I hope that it is much clear. I can check about sending my work to professional editor if needed.
>
> MOREOVER
> In order to give a stronger justification to the soft weighting mechanism I added in the appendix a section which deals with this issue.

---

### Author Response · Authors · 2018-11-06
**revision 1**

1) Clarify the novelties of the article in the abstract and in introduction too.
2) Put more related work (main part)
3) Add another experiment in section 5.3 ZERO SHOT LEARNING on bigger zero-shot dataset called 3-hops relative to the 2-hop dataset.
4) add more conclusions and future work.
5) Improve  grammar issues structure issues (moving taxonomy figure)
6) remove the ack part.


I mainly had updated this revision before have got a feedback. I am going to improve it more according the mentioned issues. Still I wanted to put this revision because I think this is much better version I still have a work to do and will work on it.
Regards

---

### Author Response · Authors · 2018-11-25
**revision 2.0, 2.1,2.2**

revision 2.0
The big difference in this revision from revision 1 is a much clear work.

- Significant English improvement and notation issues.
- Add explanations about concepts which was not clear to readers.
- Add samples which illustrates the importance of this work.
- Emphasize the the contribution of this work.
-Add more related works which deals with semantic classification and refer to state of the art zero-shot works.

revision 2.1
- Give a justification for the weighting mechanism (in appendix).

revision 2.2
-fix space issues in order to be in 10 pages limits.
- put appendix after bib. as was asked in ICLR guidelines.


next revision
- Add samples for hierarchical zero-shot.
- add the h-Correct set generation algorithm for completeness of supplementary information.
- 14.12 - I improved grammar English issues. If needed as mentioned I will be able to send the article to professional editor.

---

### Meta-Review · Area_Chair1 · 2018-12-14

**Confidence:** 4
**Recommendation:** Reject

**Metareview:**

The paper proposes to take into accunt the label structure for classification
tasks, instead of a flat N-way softmax. This also lead to a zero-shot setting
to consider novel classes. Reviewers point to a lack of reference to prior
work and comparisons. Authors have tried to justify their choices, but the
overall sentiment is that it lacks novelty with respect to previous approaches.
All reviewers recommend to reject, and so do I.